# Synthesis of polymers with on-demand sequence structures via dually switchable and interconvertible polymerizations

Ze Zhang[1], Tian-You Zeng[1], Lei Xia[1], Chun-Yan Hong[1], De-Cheng Wu[2] & Ye-Zi You[1]

The synthesis of polymers with on-demand sequence structures is very important not only for academic researchers but also for industry. However, despite the existing polymerization techniques, it is still difficult to achieve copolymer chains with on-demand sequence structures. Here we report a dually switchable and controlled interconvertible polymerization system; in this system, two distinct orthogonal polymerizations can be selectively switched ON/OFF independent of each other and they can be interconverted promptly and quantitatively according to external stimuli. Thus, the external stimuli can manipulate the insertion of distinct monomers into the resulting copolymer chains temporally, spatially, and orthogonally, allowing the on-demand precise arrangement of sequence structures in the resulting polymers. This dually switchable and interconvertible polymerization system provides a powerful tool for synthesizing materials that are not accessible by other polymerization methods.

[1] Hefei National Laboratory for Physical Sciences at the Microscale, CAS Key Laboratory of Soft Matter Chemistry, Department of Polymer Science and Engineering, University of Science and Technology of China, Hefei, Anhui 230026, People's Republic of China. [2] Beijing National Laboratory for Molecular Sciences, State Key Laboratory of Polymer Physics and Chemistry, Institute of Chemistry, Chinese Academy of Sciences, Beijing 100190, People's Republic of China. Correspondence and requests for materials should be addressed to C.-Y.H. (email: hongcy@ustc.edu.cn) or to D.-C.W. (email: dcwu@iccas.ac.cn) or to Y.-Z.Y. (email: yzyou@ustc.edu.cn)

The precise incorporation of monomers into the resulting polymer chains, forming sequence-controlled polymers, is a major challenge for modern polymer chemistry[1–13]. In molecular biology, the key processes are well executed by proteins and DNA—natural sequenced polymers with remarkable functions such as catalysis, molecular recognition, and data storage[1]. Precise sequence regulation of synthetic polymers may develop polymer materials with next-generation performance and functions. The first sequence-controlled polymer was prepared by sequential addition of monomers through peptide coupling or other coupling reactions during solid-supported step polymerization in 1963[14]. Recently, Liu and colleagues[9], and O'Reilly and colleagues[15] have elegantly developed a DNA-templated synthesis methodology for synthetic polymers, i.e., using DNA as a template and translating its sequence into synthetic polymers. Johnson et al. have reported the click synthesis of unimolecular macromolecules by an iterative exponential growth strategy, wherein the chain length, sequence, and stereo-configuration can be precisely defined. However, they are unable to produce high molecular weights and are generally time intensive. For chain polymerizations, the precise sequence regulation is more difficult given the intrinsic mechanism. However, despite the difficulty, Pfeifer and Lutz[16] proposed robust strategy for synthesizing sequence-controlled polymers via the timed addition of kinetically fast monomers. Sawamoto and colleagues[17,18] designed a template initiator with different functional groups, which can recognize different monomers through molecular interactions. Hawker and colleagues[6], and Hillmyer and colleagues[19] have prepared sequence-controlled polymers using ring-opening polymerization, etc. Although the developed methods have proved to be very effective for regulating the sequence structure, it is still very difficult to achieve copolymer chains with on-demand sequence structures, especially when diverse monomers are all present in one pot simultaneously; moreover, many advanced polymeric architectures (e.g., symmetrically gradient, gradient block, and other periodic copolymers) that are expected to possess special physical properties are still synthetically intractable[1,2,4,20–23]. Recently, the use of stimuli to selectively start or suspend polymerization processes has attracted considerable attention, providing a very powerful tool for regulating polymerization[20,24–34]. Such switchable polymerization methods not only allow for the synthesis of polymers with controlled

molecular weights but also provide opportunities to operate polymerizations temporally, spatially, and orthogonally[35–42]. For examples, Fors and colleagues[31], and Kamigaito and colleagues[43] reported a strategy that combines radical polymerization and cationic polymerization mechanisms in a one-pot setup, and switching the monomer selectivity provides an opportunity to control sequence of methyl acrylates and vinyl ethers or vinyl acetates. Boyer and colleagues[35], and Zhu and colleagues[44] have proposed a concept to use temperature to manipulate the polymer structure. Boyer and colleagues[35] also has reported an interesting method to prepare block copolymer by two orthogonal polymerizations using different wavelengths. Therefore, switchable polymerization will be a promising and powerful means for regulating the sequence structure of polymer chains.

Here we propose a dually switchable and controlled interconvertible polymerization system involving an anionic ring-opening polymerization (AROP), which only polymerizes cyclic monomers and a photoinduced electron transfer–reversible addition–fragmentation chain transfer (PET-RAFT) polymerization that is highly selective to vinyl monomers. Both the AROP and PET-RAFT polymerization can be selectively switched ON/OFF independent of each other using external stimuli, and a single trithiocarbonate unit can not only simultaneously control the AROP and PET-RAFT polymerization but also make them interconvert promptly and quantitatively on demand, as shown in Fig. 1. Therefore, the dually switchable polymerization system can polymerize cyclic monomers and vinyl monomers temporally, spatially, and orthogonally by heating or light irradiation, thus precisely manipulating the incorporations of monomers into the resulting polymer chains, forming copolymers with on-demand sequence structures.

## Results

**Switch ON/OFF of AROP and PET-RAFT polymerization.** Nishikubo and colleagues[45–47] found that quaternary onium salts could catalyze the AROP of cyclic thiirane monomers using thioester, thiourethane and trithiocarbonate as initiators. Here we found that the AROP of thiirane monomers using trithiocarbonate as the initiator was highly temperature dependent (Supplementary Figure 1), having a high apparent activation energy (Ea$^{app}$) of 100.87 kJ mol$^{-1}$. At high temperatures (> 50 °C) the AROP of thiiranes proceeded smoothly, whereas at 20 °C

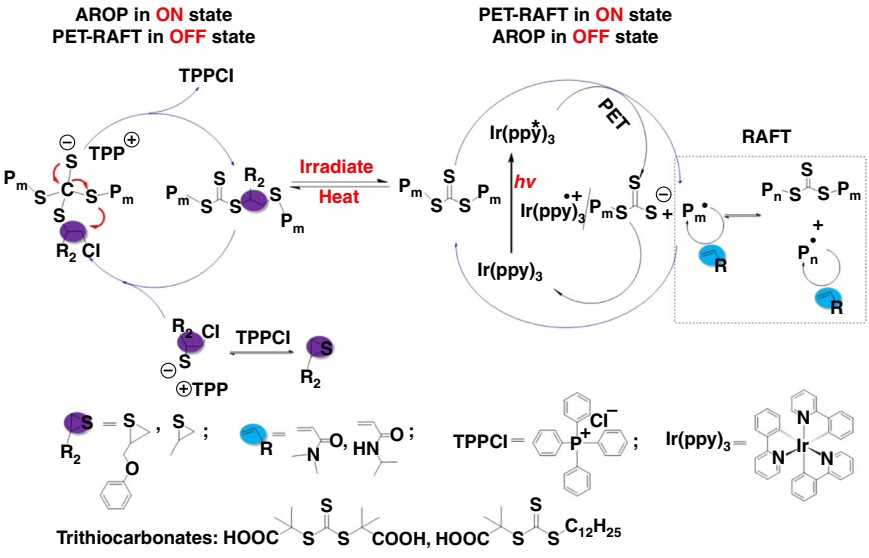

**Fig. 1** Mechanism of dually switchable and controlled interconvertible polymerizations. AROP and PET-RAFT polymerization can be switched ON/OFF and interconverted in response of stimuli

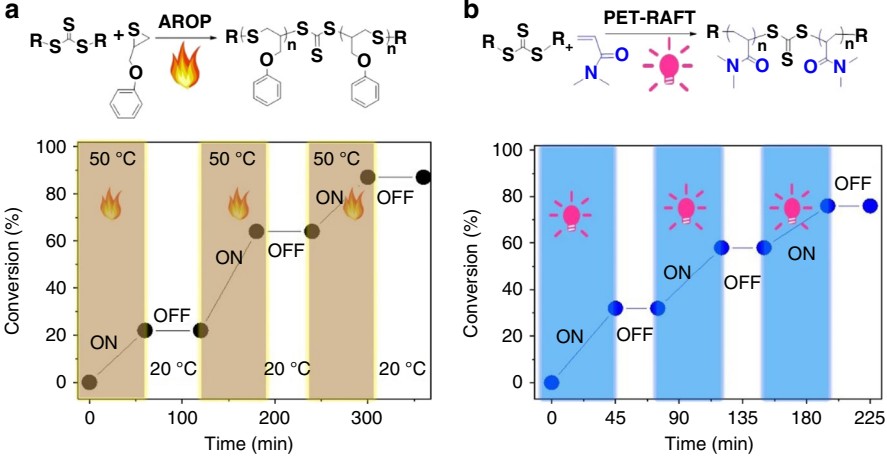

**Fig. 2** ON/OFF modes of AROP and PET-RAFT polymerization. **a** AROP and POMT conversion vs. time with and without heating. **b** PET-RAFT polymerization and DMA conversion vs. time with and without irradiation

the polymerization did not occur even after 48 h, as shown in Supplementary Figure 1. This result suggested that the polymerization could be easily toggled between ON and OFF by heating and cooling. The S,S′-bis(α,α′-dimethyl-α″-acetic acid) trithiocarbonate (BDMAT)-mediated AROP of 2-(phenoxymethyl) thiirane (POMT) in the presence of quaternary onium salts was performed at 50 °C, as shown in Fig. 2a. After 1 h of reaction, the POMT conversion reached 21%. However, when the reaction temperature quickly decreased to room temperature (20 °C), the BDMAT-mediated AROP stopped immediately. Subsequently, the polymerization could restart by increasing the reaction temperature to 50 °C and could be suspended by rapid cooling. At each polymerization step, the produced polymers had controlled molecular weights and narrow polydispersity (Supplementary Figure 2). Therefore, the AROP of thiiranes could be easily switched between ON and OFF via heating and cooling, as shown Fig. 2a. This reason is that BDMAT-mediated AROP involved the slow formation of alkyl thiolate catalyzed by quaternary onium salts, as well as a fast attack of alkyl thiolate to trithiocarbonate followed by rearrangement. The formation of alkyl thiolate is a reversible process and higher temperatures favored the catalyst to overcome the energy barrier, thus opening the ring of the thiiranes.

On the other hand, trithiocarbonates are efficient chain transfer agents to control RAFT polymerization. Recently, Boyer and colleagues[29,30,35,44,48–56] have developed a PET-RAFT polymerization method using a series of photocatalysts in response to different wavelength to control the free radical polymerization of vinyl monomers. The PET-RAFT polymerization of N,N-dimethylacrylamide (DMA) occurred under irradiation and stopped as soon as the irradiation was removed. Subsequently, the polymerization could be restarted rapidly by applying irradiation again. Therefore, the BDMAT-mediated PET-RAFT polymerization could be easily switched between ON and OFF, as shown in Fig. 2b.

**Synthesis of ABA and BAB triblock copolymers in one pot**. If the photocatalyst, quaternary onium salt, trithiocarbonate, vinyl monomer, and cyclic thiirane monomer were present in one pot simultaneously, neither the PET-RAFT polymerization of the vinyl monomer nor the AROP of the cyclic thiirane monomer occurred in the absence of irradiation or heating. When the polymerization system was irradiated with light, the photocatalyst was activated to polymerize vinyl monomers via PET-RAFT,

whereas the AROP of thiirane monomers was dormant. On removing the irradiation and heating the polymerization system, the AROP was switched on, while the PET-RAFT polymerization was switched OFF. Further, trithiocarbonate unit could be switched to mediate the AROP from mediating the PET-RAFT polymerization when the AROP was started and the PET-RAFT polymerization was stopped. In addition, it could be switched back to regulate PET-RAFT polymerization when the heating was stopped and irradiation was applied to the system. Thus, the external stimuli could not only switch both AROP and PET-RAFT polymerization between ON and OFF but also make the polymerizations interconvert easily.

To verify the hypothesis, we chose Ir(ppy)₃ as the photocatalyst for the PET-RAFT polymerization and tetraphenylphosphonium chloride (TPPCl) as the catalyst for the AROP (see experiment details in Supplementary Methods). When Ir(ppy)₃, TPPCl, BDMAT, DMA, and POMT were all present at once, we first subjected the polymerization system to heating followed by irradiation (Fig. 3a). Under heating without irradiation, the AROP was switched ON, as shown in Fig. 3b. The POMT conversion reached 12% and 72% in 1 and 4.5 h (Supplementary Figures 3 and 4), respectively. A linear increase in $\ln([M]_0/[M])$ with reaction time was observed (Fig. 3c). In the UV–vis spectra, the absorption peak of trithiocarbonate shifted to 438 from 453 nm, because its neighboring units changed into POMT units (see Supplementary Figure 6). We used $^1$H–nuclear magnetic resonance (NMR) to trace the polymerization. It was very obvious that the signals attributed to polymer of POMT (PPOMT) gradually strengthened with increasing heating time, and that no peaks corresponding to polymer of DMA (PDMA) were present (Fig. 3f and Supplementary Figures 3 and 4), indicating that none of the DMA was consumed under heating. The polymers obtained under heating showed unimodal size exclusion chromatography (SEC) curves and controlled molecular weights (Fig. 3d, e). Subsequently, the mixture was rapidly cooled to room temperature and exposed to a 5-W blue light-emitting diode (LED). $^1$H-NMR results showed that the AROP of POMT stopped as soon as the heating was suspended. Under irradiation, Ir(ppy)₃ was activated to trigger the PET-RAFT polymerization of DMA, giving a 62% monomer conversion after 5 h irradiation (Fig. 3b and Supplementary Figure 5). For the PET-RAFT polymerization, a linear relationship between $\ln([M]_0/[M])$ and reaction time, and between $M_n$ and DMA conversion was observed, illustrating that this polymerization could be well controlled using trithiocarbonate-containing PPOMT as the

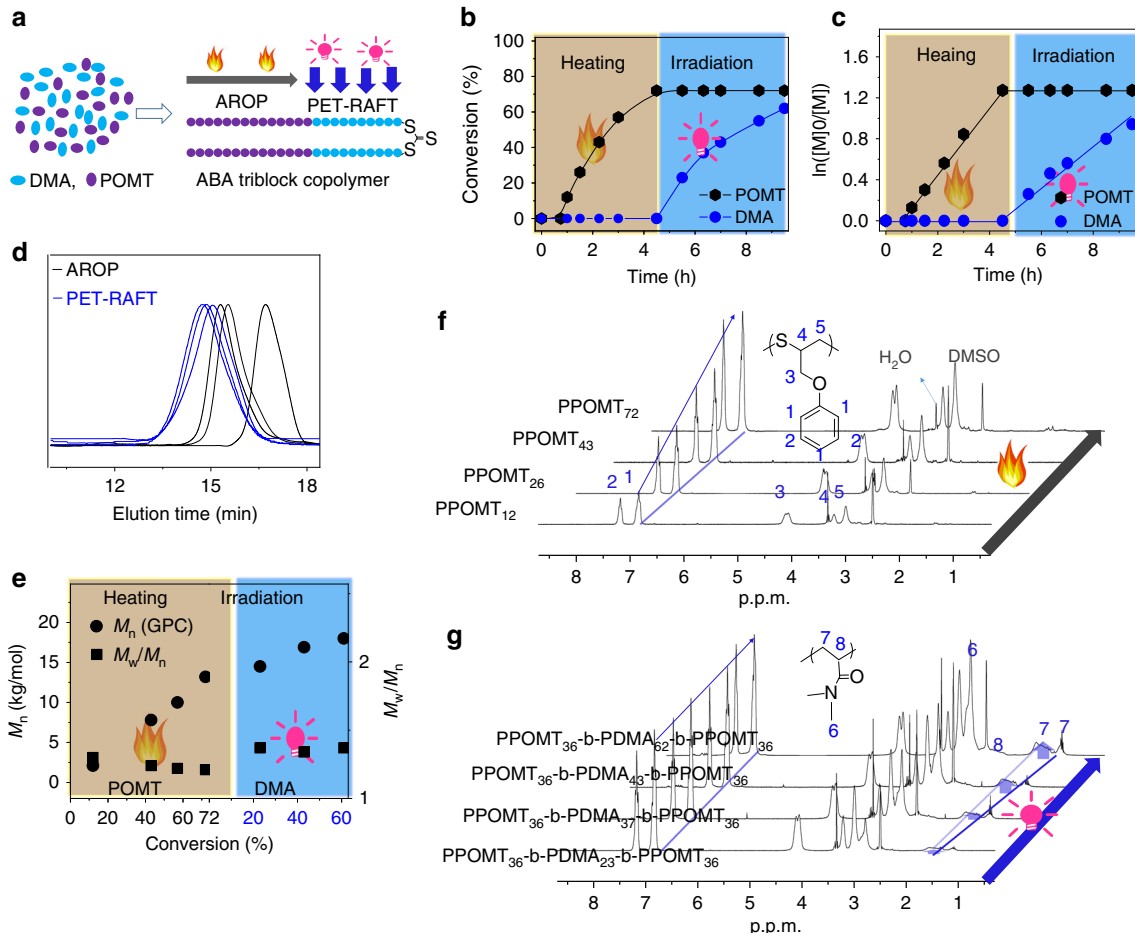

**Fig. 3** Synthesis of ABA triblock copolymer by heating followed by irradiation. **a** Scheme of synthesis of ABA triblock copolymer. **b** Monomer conversion versus polymerization time. **c** $\ln([M]_0/[M])$ vs. reaction time. **d** SEC curves for resulting polymers at different polymerization times under heating followed by irradiation. **e** Molecular weight and polydispersity versus monomer conversion under heating followed by irradiation. **f** $^1$H-NMR spectra of the polymers produced at different conversion under heating. **g** $^1$H-NMR spectra of the polymers produced at different conversion under heating followed by irradiation

macro-RAFT agent (Fig. 3c). Based on the $^1$H-NMR spectra, none of POMT was consumed under irradiation. In the UV–vis spectra, the absorption peak of trithiocarbonate shifted from 438 to 433 nm, because the neighboring units of trithiocarbonate changed from POMT into DMA units (Supplementary Figure 6). In addition, the proton signal intensity of PPOMT remained unchanged under irradiation, whereas the signal intensity of DMA appeared and strengthened with increasing the irradiation time (Fig. 3g), forming an ABA-type triblock copolymer containing two PPOMT segments and one PDMA segment with average 62 DMA units and 72 POMT units in one polymer chain. The Fourier infrared (FI-IR) spectra were also consistent with NMR analysis (Supplementary Figure 7). The above results indicated that the AROP and PET-RAFT polymerizations could be switched ON or OFF independent of each other, and that the AROP could be easily converted into the PET-RAFT polymerization when the external stimulus was changed from heating to irradiation. Further, the AROP and PET-RAFT polymerization showed highly selective monomer discrimination, which allowed a single propagating chain to selectively polymerize cyclic monomers and vinyl monomers on demand.

Second, we subjected the polymerization system to irradiation followed by heating. Under irradiation without heating, the PET-RAFT polymerization was activated and DMA started to polymerize. As irradiation time increased, the signal intensity of the peak corresponding to DMA and $M_n$ value of the produced

PDMA gradually increased and no peaks were observed for POMT (Supplementary Figures 8, 9, and 10). Subsequently, after the external stimulus was changed from irradiation to heating, the AROP of POMT was activated, whereas the PET-RAFT polymerization of DMA was suspended immediately (Supplementary Figures 8, 9, and 11). As heating time increased, the signals corresponding to POMT became stronger. The linear plots of $\ln([M]_0/[M])$ as a function of reaction time and $M_n$ as a function of monomer conversion were also observed for both heating and irradiation (Supplementary Figure 8). According to $^1$H-NMR results, on an average, 77 DMA units and 74 POMT units were copolymerized into the polymer chain with a BAB triblock structure (Supplementary Figure 9). The above results demonstrated that there was no interference between the PET-RAFT and AROP, and that they could be interconverted promptly and quantitatively according to the external stimulus (heating or irradiation). Further, the polymerization could be switched back and forth on demand between AROP and PET-RAFT polymerization based on on demand by changing the external stimulus. The unique characteristics of this polymerization strategy will provide more powerful and precise controls of the polymer sequence structure.

**Synthesis of multiblock copolymers in one pot.** Multiblock copolymers possess special physical properties compared to common block copolymers[23,57], but their synthesis is

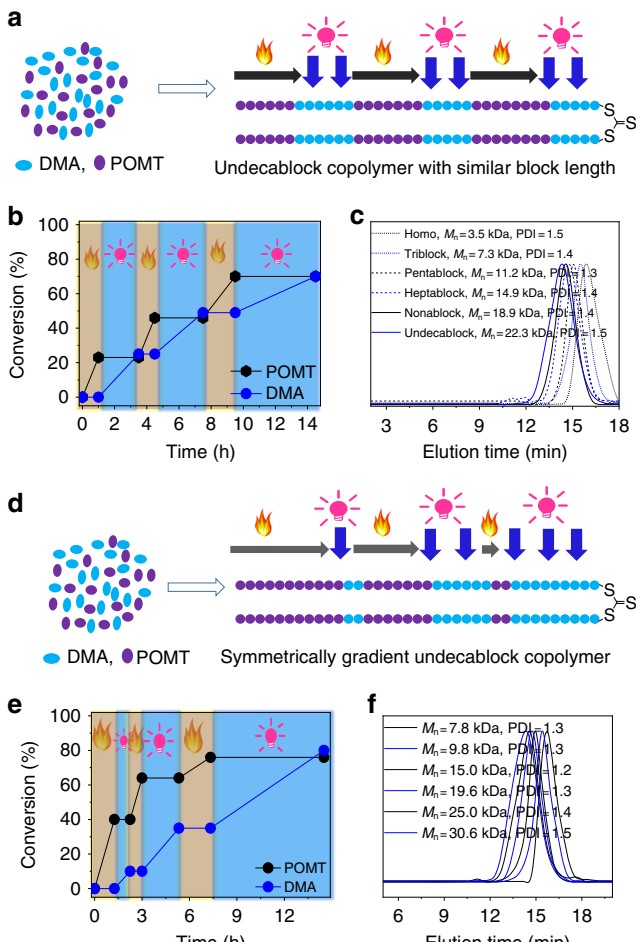

**Fig. 4** Synthesis of multiblock copolymer by multiple cycles of heating followed by irradiation. **a** Outline of formation of undecablock copolymer with similar block length. **b** Monomer conversion versus polymerization time for synthesis of undecablock copolymer with similar block. **c** SEC curves for resulting polymers at different polymerization time. **d** Outline of formation of symmetrically gradient undecablock copolymer. **e** Monomer conversion versus polymerization time for synthesis of symmetrically gradient undecablock copolymer. **f** SEC curves for resulting polymers at different polymerization time

synthetically intractable[2]. Based on the above dually switchable system, multiblock copolymers could be prepared simply by subjecting the polymerization system to multiple rounds of heating followed by irradiation, such that vinyl and cyclic monomers were all present in one pot at once (Fig. 4a), e.g., the polymerization of DMA and POMT in the presence of BDMAT, TPPCl, and Ir(ppy)$_3$ (see experiment details in Supplementary Methods). First, the AROP of POMT was started by heating the polymerization system to 60 °C and 23% of the POMT was converted to PPOMT with a trithiocarbonate unit in the middle of the PPOMT chain after 1 h, forming homo-PPOMT with $M_n = 3.5$ kDa (Fig. 4b, c and Supplementary Figure 12). After quickly cooling the system to room temperature and exposing it to a blue LED, the PET-RAFT polymerization of DMA was activated, while the AROP of POMT was suspended. Up to 25% of the DMA monomer was consumed in 2.5 h, forming a triblock copolymer containing two PPOMT blocks on either side of a DMA block, with $M_n = 7.3$ kDa (Fig. 4b, c and Supplementary Figure 12). Subsequently, the AROP of POMT was restarted by applying heat and removing the irradiation and another 22% of the POMT monomer was inserted into the

block copolymer chain in 0.67 h, forming a pentablock copolymer having three PPOMT segments and two PDMA segments, with $M_n = 11.2$ kDa. Then, multiple rounds of heating followed by irradiation were alternatively applied continuously, to from heptablock, nonablock, and undecablock copolymers (Fig. 4b, c and Supplementary Figure 12). All the copolymers showed unimodal SEC curves and had controlled molecular weights. After three cycles of the PET-RAFT polymerization and AROP, an undecablock copolymer with six PPOMT blocks and five PDMA blocks was formed. Due to the similar conversion of both monomers during the cycles, each block had a similar length. Each block of PPOMT included ~ 14 units, whereas each block of PDMA included ~ 15 units (Supplementary Figure 12). During the synthesis of multiblock copolymers, the orthogonality of AROP and PET-RAFT was also confirmed by a controlled experiment (Supplementary Figures 13 and 14). To further verify the formation of the produced multiblock copolymers via this switchable polymerization system, we performed DSC analysis of the resulting undecablock copolymer. Generally, the $T_g$ of PPOMT is 9 °C, whereas that of PDMA is 112 °C (Supplementary Figure 15). In DSC curve of the ABA triblock copolymer, two $T_g$s were observed (Supplementary Figure 16a), demonstrating microphase separation of long blocks in the resulting copolymer. Interestingly, the DSC curve of the resulting undecablock copolymer had only one broad $T_g$ around 46 °C (Supplementary Figure 16b), which may have resulted from that no ordered structures or nanophase separation of nearly pure PPOMT or nearly pure PDMA segments was formed due to the short block length of PPOM or PDMA in the resulting polymer.

In this dually switchable and interconvertible polymerization system, the block numbers of the copolymer could be controlled by varying the number of cycles: one cycle resulted in the formation of a triblock copolymer, and two, three, four, and five cycles led to the formation of pentablock, heptablock, nonablock, and undecablock copolymers, respectively. The chain length for each block could be easily regulated by the irradiation and heating time. Thus, by simply varying the irradiation and heating time, a symmetrically gradient block copolymer (PPOMT$_{30}$-b-PDMA$_8$-b-PPOMT$_{18}$-b-PDMA$_{19}$-b-PPOMT$_9$-b-PDMA$_{34}$-**SC( = S)**-PDMA$_{34}$-b-PPOMT$_9$-b-PDMA$_{19}$-b-PPOMT$_{18}$-b-PDMA$_8$-b-PPOMT$_{30}$), which has so far been synthetically intractable, was easily obtained (Fig. 4d–f). For this long copolymer chain, the PPOMT blocks at both ends were the longest (30 POMT units) and the gradient decreased toward the middle (9 POMT units); further, the length of the DMA block gradient decreased from the middle (68 DMA units) to the ends (8 DMA unit). In the DSC curves (Supplementary Figure 17), two distinct glass transition intervals (7–28 °C, 71–82 °C), which were close to those of the homo-polymers of PPOMT and PDMA, were observed; these were attributed to the long POMT blocks at both ends and long DMA block in the middle, respectively. Furthermore, very broad $T_g$ range between 28 °C and 67 °C was observed, which corresponded to the broad phase containing mixed short PPOMT blocks and PDMA blocks sandwiched between long PPOMT blocks and PDMA blocks. The overall glass transition temperature range was 75 °C, which was similar to those of typical gradient copolymers[58].

**Synthesis of heptablock quadricopolymer in one pot.** The dually switchable and interconvertible polymerization system showed broad monomer applicability, high efficiency, and no interference between the PET-RAFT polymerization and AROP. New monomers 2-methylthiirane (MT) and N-isopropyl acrylamide (NIPAM) were tested, and the results demonstrated that only MT was polymerized on heating, whereas only NIPAM was polymerized with blue light. Furthermore, there was no any

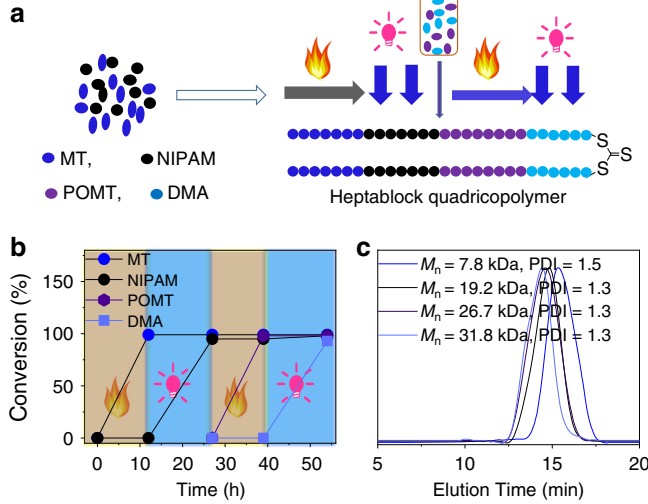

**Fig. 5** Synthesis of heptablock quadicopolymer. **a** Outline of formation of heptablock quadicopolymer. **b** Monomer conversions vs. polymerization time for synthesis of heptablock quadricopolymer. **c** SEC curves for resulting polymers at different polymerization time

consumption of both MT and NIPAM without light or heat, confirming the orthogonality of MT and NIPAM (Fig. 5a and Supplementary Figures 18 and 19). Therefore, multiblock copolymers with more diverse block structures could be easily prepared by sequential addition of comonomers and sequential heating followed by irradiation, e.g., the comonomers MT and NIPAM were first added into a polymerization tube, both MT monomer conversion and NIPAM monomer conversion reached ~ 100% after applying 12 h of heating followed by 15 h of irradiation, forming a triblock copolymer with two PMT blocks and one PNIPAM block, as shown in Fig. 5a, b; subsequently, the comonomers POMT and DMA were injected into the polymerization tube. On further heating for 12 h followed by irradiation for 15 h, a heptablock quadricopolymer with two PMT blocks, two PNIPAM blocks, two PPOMT blocks, and one PDMA block was formed (Fig. 5a,b). Based on the [1]H-NMR and [13]C–NMR spectra (Supplementary Figures 20 and 21), all four starting monomers were present in the resulting heptablock copolymer. SEC data (Fig. 5c) further demonstrated the controlled behavior during polymerization.

## Advanced sequence control by programmed heating and irradiation.
The sequences in biomacromolecules have a vital role in the development, functioning, and reproduction of all living organisms. The ability to construct highly organized sequence-controlled synthetic macromolecules would be a significant breakthrough with potential applications in many fields, including nanomedicine and nanotechnology[1,4]. However, the most developed strategies can only couple one building block at a time, and in many cases, they require protection/deprotection of the growing oligomer chains. Therefore, these strategies are expensive and time consuming, and only achieve a partial sequence control[2,3]. This dually switchable polymerization system proposed in the present study provides versatile means to precisely manipulate the sequence structure of the resulting polymers. As shown in Fig. 6a–c and Supplementary Figure 22, using the well-designed program of heating combined with intermittent irradiation, the DMA unit could be precisely inserted into the PPOMT chain by simply irradiating the polymerization tube (see experiment details in Supplementary Methods). The length of each PPOMT segment could be controlled by the heating time.

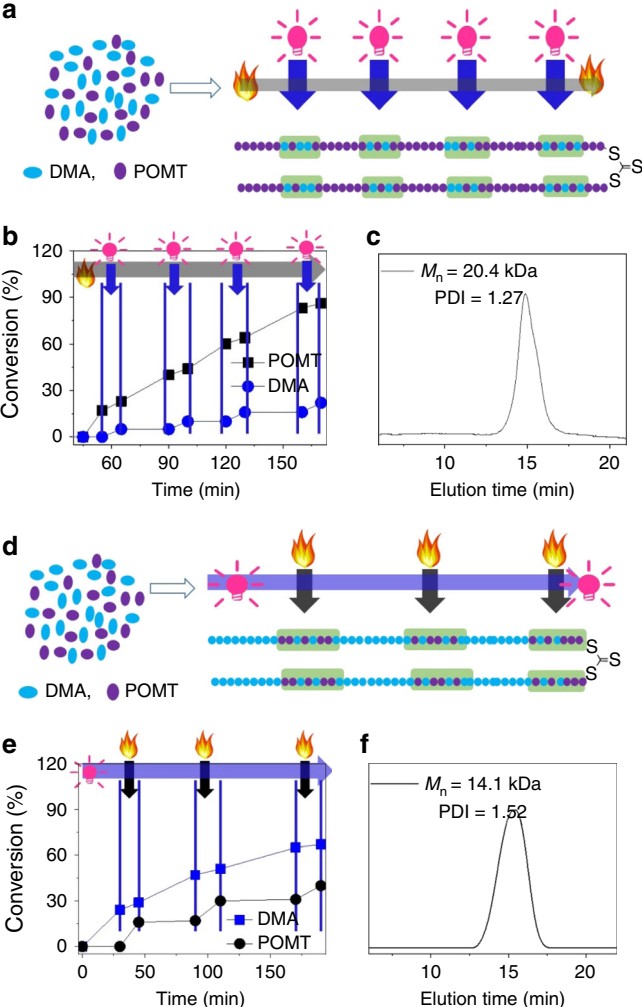

**Fig. 6** Synthesis of copolymers with controlled sequence structures. **a** Scheme of synthesis of copolymers with controlled sequence structures via continuous heating combined with intermittent irradiation. **b** Monomer conversion vs. polymerization time under continuous heating combined with intermittent irradiation. **c** SEC curve of a typical resulting copolymer with a controlled sequence structure formed via continuous heating combined with intermittent irradiation. **d** Scheme of synthesis of copolymer with controlled sequence structures via continuous irradiation combined with intermittent heating. **e** Monomer conversion vs. polymerization time under continuous irradiation combined with intermittent heating. **f** SEC curve of resulting copolymer with controlled sequence structures formed via continuous irradiation combined with intermittent heating

The location of the DMA unit in the PPOMT chain could be tuned by the irradiation time. The amount of the DMA and the length of the DMA and POMT in the co-segment could be regulated by the irradiation time and amount of Ir(ppy)₃. Similarly, using a tailored program of continuous irradiation combined with intermittent heating, the POMT unit could be precisely inserted into the PDMA chain by simply heating the polymerization tube. The position of the POMT units in the PDMA chain could also be manipulated by the heating time; the number of the DMA units could be controlled by the heating time and amount of TPPCl added (Fig. 6d–f and Supplementary Figure 23). Therefore, we could precisely insert the DMA units into the PPOMT chain at any position simply via irradiation and insert the POMT units into the PDMA chain at any position simply by heating (Fig. 6).

Moreover, the proposed method is versatile, simple, and flexible. The program of heating and irradiation could be well-designed on demand, and hence the resulting copolymers with advanced sequence structures could be obtained. The sequence structure depends on the designed program of heating and irradiation and not on reactivity of the monomers. Therefore, to synthesize triblock copolymer PPOMT block-*b*-copolymer block-*b*-PDMA block, we should use the program of sequential heating, heating integrated with irradiation, and irradiation on the polymerization system, whereas to synthesize the triblock copolymer PPOMT block-*b*-PDMA block-*b*-copolymer block, we should use the program of sequential heating, irradiation, and heating integrated with irradiation on the polymerization system (see experiment details in Supplementary Methods). As shown in Fig. 7a–c and Supplementary Figure 24, the triblock copolymer

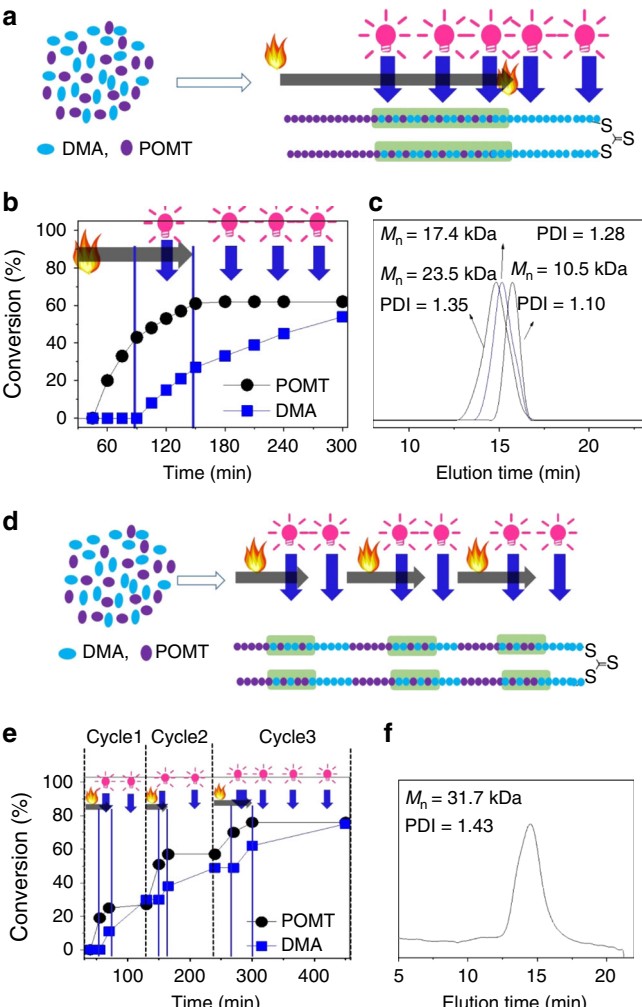

**Fig. 7** Synthesis of copolymers with advanced sequence structures. **a** Scheme of synthesis of copolymers with controlled sequence structures via sequential heating, heating integrated with irradiation, and irradiation. **b** Monomer conversion vs. polymerization time under sequential heating, heating integrated with irradiation, and irradiation. **c** SEC traces of polymer formed under sequential heating, heating integrated with irradiation, and irradiation. **d** Scheme of synthesis of copolymers with advanced sequence structures via multiple cycles of sequential heating, heating integrated with irradiation, and irradiation. **e** Monomer conversion vs. polymerization time under multiple cycles of sequential heating, heating integrated with irradiation, and irradiation. **f** SEC trace of a typical polymer formed under multiple cycles of sequential heating, heating integrated with irradiation, and irradiation

PPOMT block-*b*-copolymer block-*b*-PDMA block was obtained when the program of sequential heating, heating integrated with irradiation, and irradiation was applied to the polymerization system. As shown in Fig. 7d–f and Supplementary Figure 25, the copolymers had multiple blocks of PPOMT, PDMA, and copolymer of POMT and DMA when multiple rounds of heating, heating integrated with irradiation, and irradiation were applied to the polymerization system. The location and length of the PPOMT blocks, PDMA blocks, and copolymer blocks of POMT and DMA could be well tuned.

## Discussion

We design a dually switchable and controlled interconvertible polymerization system involving the AROP and PET-RAFT polymerization to obtain sophisticated microstructures; in this system, the AROP and PET-RAFT polymerization could be reversibly activated/inactivated rapidly and quantitatively according to the external stimuli. Further, the AROP and PET-RAFT polymerization could be fast interconverted and they show a highly selective monomer discrimination. Therefore, the well-designed program of heating and irradiation results in the synthesis of copolymers with a tailored composition, chain length, and on-demand sequence structure. Moreover, many other functional dithioesters and trithiocarbonates are also found to be suitable for this polymerization system, as well as diverse functional monomers; thus, a wide range of functionalities along the copolymer backbones with controlled physico-chemical properties could be incorporated in this system, leading to the formation of highly ordered materials with unique functions and properties. Therefore, the dually switchable and interconvertible polymerization system paves the way for the synthesis of a new class of macromolecules with on-demand sequence structures for a wide range of applications including nanostructured materials, polymeric phase separation, single-chain folding, and drug delivery.

## Methods

**Polymerization via heating followed by irradiation**. DMA (891 mg, 9 mmol), POMT (1494 mg, 9 mmol), BDMAT (25.5 mg, 0.09 mmol), photocatalyst (27 μL, 15 p.p.m.), and TPPCl (11 mg, 0.03 mmol) were dissolved in DMAc to obtain 6 mL solution, which was further transferred into a glass tube with rubber plug. After three freeze–pump–thaw cycles, the tube was sealed and immersed in an oil bath at 60 °C. After suitable time interval, polymerization solution was taken out by injector. Conversions of POMT were calculated by [1]H-NMR spectra. After 4.5 h heating (conversion of POMT reaching 72%), the tube was quickly cooled to room temperature and irradiated by 5 W blue LED strip. Similarly, after suitable time interval, polymerization solution was taken out by injector and conversions of DMA were calculated by [1]H-NMR spectra. All the reaction mixtures at different reaction time were precipitated into diethyl ether several times and the products as light yellow solids were obtained after dried in vacuum.

**Polymerization via irradiation followed by heating**. DMA (891 mg, 9 mmol), POMT (1494 mg, 9 mmol), BDMAT (25.5 mg, 0.09 mmol), photocatalyst (27 μL, 15 p.p.m.), and TPPCl (11 mg, 0.03 mmol) were dissolved in DMAc to obtain 6 mL solution, which was further transferred into a glass tube with rubber plug. After three freeze–pump–thaw cycles, the tube was sealed and irradiated by 5 W blue LED strip. After suitable time interval, polymerization solution was taken out by injector and the conversions of DMA were calculated by [1]H-NMR spectra. After 2.5 h irradiation (conversion of DMA reaching 77%), turn off the light source and the tube was immersed in an oil bath at 60 °C. Similarly, after suitable time interval, polymerization solution was taken out by injector and the conversions of POMT were calculated by [1]H-NMR spectra. All the reaction mixtures at different reaction times were precipitated into diethyl ether several times and the products as light yellow solids were obtained after dried in vacuum.

**Synthesis of multiblock copolymers**. DMA (891 mg, 9 mmol), POMT (1494 mg, 9 mmol), BDMAT (20.1 mg, 0.072 mmol), photocatalyst (45 μL, 25 p.p.m.), and TPPCl (11 mg, 0.03 mmol) were dissolved in DMAc to obtain 6 mL solution, which was further transferred into a glass tube with rubber plug. After three freeze–pump–thaw cycles, the tube was sealed and immersed in an oil bath at 60 °C. After conversion of POMT reaching 23%, the tube was quickly cooled to

room temperature and irradiated by 5 W blue LED strip. Similarly, after conversion of DMA reaching 25%, the light was turned off and the tube was immersed into oil bath at 60 °C again. After conversion of POMT reaching 46%, the tube was cooled to room temperature and irradiated by blue LED strip again. Then after conversion of DMA reaching 49%, the light was turned off and the tube was immersed into oil bath at 60 °C for the third time. After conversion of POMT reaching 70%, the tube was cooled to room temperature and irradiated by blue LED strip. All the reaction mixtures at different reaction time were precipitated into diethyl ether several times and the products as light yellow solids were obtained after dried in vacuum.

**Data availability**. The data that support the findings of this study are available within the article and its Supplementary Information File or from the corresponding author upon reasonable request.

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

## Acknowledgements

We acknowledge funding support from the National Key R&D Program of China (2017YFA0205601), the National Natural Science Foundation of China (Grant Numbers 51625305, 21704095, 21774113, and 21525420) and the Foundamental Research Funds for the Central Universities (WK9110000004).

## Author contributions

Y.Z.Y. and C.Y.H. conceived the project and directed the experiment. Z.Z. conducted the polymerizations and polymer characterization, and L.X., T.Y.Z., D.C.W., and C.Y.H. performed the data analysis. Z.Z., C.Y.H., D.C.W., and Y.Z.Y. wrote the paper. All the authors discussed the results and contributed to the manuscript.

## Additional information

**Competing interests:** The authors declare no competing interests.

