## [Peer Review File · Nature Communications]

Reviewers' comments:

Reviewer #1 (Remarks to the Author):

This manuscript reports the use of heat and light as external stimuli to access sequence-controlled polymers by successively switching between two polymerization routes, i.e AROP and PET-RAFT. Different block(y) polymers with various compositions of monomers have been synthesized and mainly characterized by DSC.

Straightforward synthesis of sequence-controlled polymers is challenging and any new methodologies are of high interest in the field. This paper combines two well-known polymerisations, i.e AROP and PET-RAFT exploiting some key features of both routes. As mentioned by the authors, using a multi-stimuli system (heat and light) and a one-pot approach (with all the monomer present at the beginning) is very attractive. In this vein, it should be noted that Johnson and coworkers recently reported a Logic-Controlled Radical Polymerization with Heat and Light (J.Am. Chem. Soc. 2017, 139, 2257-2266).

All in all, this work features an interesting approach towards tailor-made materials. However, too many data and characterizations are missing to fully support the conclusion. The formation of (multi) block polymers and gradient block polymers is not fully addressed. Lots of effort has been made to make nice figure easy to understand and to follow (and it is) However, more data (NMR spectrum, GPC traces...) should be reported in the ESI to convince the reader of the orthogonally and the formation of the claimed polymers.

Thus, the following points need to be clarify:

A draw with the structure of monomers and initiators would help the reader to follow.

Line 27: "or other external stimuli". Ref needed (for example 45)

Line 81: it should be Fig2 not Fig. 1

Line 106-119: it is not clear that is the envisioned procedure. It should be clearly stated.

NMR spectra showing the ON/OFF switch should be reported in the ESI, clearly annotated and showing unreacted monomer and formation of polymers to convince the reader. The chemical shift used to calculate conversion should be specified. For example, line 134 for 12% and 72%, and line 151 "based on ¹H NMR spectra"

Generally, the stacked NMR spectra are not easy to read.

Is it possible to observe by NMR the linkage between two blocks?

For example, NMR spectra of isolated polymers obtained using a mixture but only with light or heat and featuring only one type of enchainment would be convincing of the orthogonality.

Line 158: 62 DMA and 72 POMT: how is it calculated/estimated?

Figure 3d: how to explain the PDI values decreasing with higher conversion and formation of block copolymers?

Test reaction: after a reversible switch, what happen in the absence of both light and heat? Does the polymerisation completely stop? is it possible to restart ?

Figure 4f: PPDI values are missing

Line 267: new monomers are tested but the orthogonality has not been proved (as it has been done with DMA and POMT)

Line 268-273: Clearly annotated NMR spectra are needing.

How different are the NMR spectra corresponding to block copolymer to gradient polymers (obtained by light and heat at the same time). New linkage should be observable by NMR.

Reviewer #2 (Remarks to the Author):

This is an excellent paper. The experimental data support the conclusion of this paper. Although there are few examples of orthogonal polymerizations, see ROP and PET-RAFT was perhaps the first one using two different wavelength (Chemical Communications 52, 7126-7129), this paper demonstrates two different stimuli, temperature and light. Both stimuli appears orthogonal. In my opinion, this paper meets the criteria for publication in Nat. Comm.

I have some minor comments:

1. the authors should comment and explain what is the difference between reference 30 and this one. Indeed, ref 30 demonstrates two orthogonal polymerization controlled by 2 different wavelengths. This is relatively close to this work. As a consequence, the authors should clearly describe this previous work.
2. I recommend to include NMR data in the main text. Indeed, it is important information to provide to the readers to demonstrate that both monomers are copolymerized under a specific stimuli.
3. The authors provide numerous schemes in the text. It is obviously good to illustrate the text with schemes, but in my opinion, too many schemes are used. I recommend to incorporate more experimental data, i.e. SEC traces, and NMR to support these conclusions.
4. The concept to use temperature to manipulate the polymer structure has also been proposed in this paper: Polymer Chemistry 8 (39), 6024-6027.
5. Some sentences need to be improved.
6. DSC experiments, I fully agree with the authors that one Tg could mean two monomers are copolymerized. can the authors add a control experiment showing that if these copolymerize in sequence, i.e. prepare block copolymers, two Tg will be observed.
7. perhaps, the author could be interested to cite this paper: Chemical Science 6 (2), 1341-1349 in the introduction showing that PET-RAFT process can use a wide range of catalysts.

Overall, I strongly recommend publication of this paper.

Reviewer #3 (Remarks to the Author):

This is a nice paper describing what in essence is a very simple idea which is reactions that occur at elevated temperatures will have a very slow rate at ambient temperature. Reactions that require light can usually occur at ambient temperature. Thus simply by lowering the temperature reactions that require 60-90C can pretty much be stopped whilst light is applied. Having said that I am unaware of anyone else who has applied this simple premise and thus this is simplicity is the papers main aspect. They have chosen a rather esoteric AROP monomer rather than a more simple reaction but this is obviously something they are familiar with so why not. They go on to use this to make a multiblock copolymer and show ABA can be made BAB. There is a question over the applications of multi-block copolymers and so it is good to see the collection of thermal data as a measure of properties. However, this is the least convincing of the data and I would like to see maybe DMA or a better measure of Tg prior to acceptance.

It is good to see the GPC traces but I would like to see the data collated into a table. Also we see considerable broadening of the dispersity. What is the dispersity of the polymers shown in suppl figure 8 for example. We are told in the text the curves are all unimodal but they look quite broad so how do the authors know this?

Then maybe a discussion of why dispersity seems to broaden as blocks are added.

Dear Referees:

On behalf of my co-authors, we thank you very much for your very helpful comments concerning our manuscript entitled **“Synthesis of Polymers with On-Demand Sequence Structures via Dually Switchable and Interconvertible Polymerizations” (NCOMMS-18-03191)**. Those comments are very valuable and very helpful for revising and improving our paper, as well as the important guiding significance to our researches. We have made corrections, which we hope meet with approval.

Thank you and best regards.

Responds to the reviewers' comments:

Reviewer #1:

This manuscript reports the use of heat and light as external stimuli to access sequence-controlled polymers by successively switching between two polymerization routes, i.e AROP and PET-RAFT. Different block(y) polymers with various compositions of monomers have been synthesized and mainly characterized by DSC.

Straightforward synthesis of sequence-controlled polymers is challenging and any new methodologies are of high interest in the field. This paper combines two well-known polymerisation, i.e AROP and PET-RAFT exploiting some key features of both routes. As mentioned by the authors, using a multi-stimuli system (heat and light) and a one-pot approach (with all the monomer present at the beginning) is very attractive. In this vein, it should be noted that Johnson and coworkers recently reported a Logic-Controlled Radical Polymerization with Heat and Light (J.Am. Chem. Soc. 2017, 139, 2257-2266) (We have cited this paper in our revised manuscript).

All in all, this work features an interesting approach towards tailor-made materials. However, too many data and characterizations are missing to fully support the conclusion. The formation of (multi) block polymers and gradient block polymers is not fully addressed. Lots of effort has been made to make nice figure easy to understand and to follow (and it is) However, more data (NMR spectrum, GPC traces...) should be reported in the ESI to convince the reader of the orthogonally and the formation of the claimed polymers.

The following points need to be clarify:

1. A draw with the structure of monomers and initiators would help the reader to follow.

Reply: We thank the Referee for this very valuable suggestion. **A new draw with the structures of monomers, initiators and catalysts** has been added in the revised manuscript as **Scheme 1**.

2. Line 27: “or other external stimuli”. Ref needed (for example 45), Line 81: it should be Fig.2 not Fig.1

Reply: Thank the referee very much for the very valuable suggest. We have added a review on “external regulation on controlled polymerization (**Angew. Chem. Int. Ed. 2013, 52, 199**) and **Ref 45** as references here (as **Ref. 2** and **Ref 16**). Very sorry, we have made mistake, it should be Fig.2 not Fig.1, and we have revised it.

3. Line 106-119: it is not clear that is the envisioned procedure. It should be clearly stated.

Reply: We thank the Referee for this valuable suggestion. We have modified corresponding statement to make it clear in our revised manuscript as follows:

“Therefore, if **the** photocatalyst, quaternary onium salt, trithiocarbonate, vinyl monomer and cyclic thiirane monomer **were** present in one pot **simultaneously**, neither **the** PET-RAFT polymerization of **the** vinyl monomer nor the AROP of **the** cyclic thiirane monomer occurred **in the absence of** irradiation or heating. **When the** polymerization system **was irradiated with**

light, the photocatalyst was activated to polymerize vinyl monomers via the PET-RAFT polymerization, while the AROP of thiirane monomers was dormant. On removing the irradiation and heating the polymerization system, the AROP was switched on, while the PET-RAFT polymerization was switched OFF. Further, trithiocarbonate unit could be switched to mediate the AROP from mediating the PET-RAFT polymerization when the AROP was started and the PET-RAFT polymerization was stopped. Also, it could be switched back to regulate PET-RAFT polymerization when the heating was stopped and irradiation was applied to the system. Thus, the external stimuli could not only switch both AROP and PET-RAFT polymerization between ON and OFF but also make the polymerizations interconvert easily.”

4. NMR spectra showing the ON/OFF switch should be reported in the ESI, clearly annotated and showing unreacted monomer and formation of polymers to convince the reader. The chemical shift used to calculate conversion should be specified. For example, line 134 for 12% and 72%, and line 151 “based on ^1H NMR spectra” Generally, the stacked NMR spectra are not easy to read.

Reply: We thank the Referee very much for the very valuable suggest. Conversions were recorded by ^1H NMR spectra after suitable time interval with NMR tubes adapted with coaxial inserts. D_2O was in the inner of the concentric capillary tube, while the mixed solution (solvent: DMAc) in the outer capillary tube. NMR spectra showing the ON/OFF switch during the formation of ABA and BAB triblock copolymers have been reported in the ESI (**Supplementary Figure 3 and Supplementary Figure 6**), clearly annotated and showing unreacted monomer and formation of polymers. The chemical shift used to calculate conversion and the calculation method have also been specified, as shown in **Supplementary Figure 3, Supplementary Figure 6**. It is very true that the stacked NMR spectra are not easy to read, in **Supplementary Figure 3, Supplementary Figure 6**, the NMR spectra were separated, not stacked. In Fig 2e and 3e, some NMR spectra were stacked, we wanted to show the comparable changes of the signals of PDMA or PPOMT during the polymerization.

5. Is it possible to observe by NMR the linkage between two blocks? For example, NMR spectra of isolated polymers obtained using a mixture but only with light or heat and featuring only one type of enchainment would be convincing of the orthogonality.

Reply: 1) Thank the Referee for the valuable suggest, we have run the experiment and observed that NMR spectra of isolated polymers obtained using a mixture but only with light or heat have shown only one type of enchainment. For examples, NMR spectra of isolated polymers obtained using a mixture of DMA and POMT with only heating exhibited that peaks of PPOMT increased as time increased and that there was no any signals of PDMA(**Fig. 2e, Supplementary Figure 3**), while NMR spectra of isolated polymers obtained using a mixture of DMA and POMT with only irradiation showed that peaks of PDMA increased as time increased and that there was no any signals of PPOMT(**Fig. 3e, Supplementary Figure 6**).

2) **It is possible to observe by NMR the linkage between two blocks**, for examples, the block copolymer of MT and DMA, when using irradiation followed by heating, PDMA-*b*-PMT-*b*-PDMA triblock polymer was obtained, and we can observe proton signal of linkage between PDMA segment and PMT segment (**1**).

When using heating followed by irradiation, PMT-*b*-PDMA-*b*-PMT triblock polymer was obtained, and we can observe proton signal of linkage between PMT segment and PDMA segment (2).

3) Furthermore, in carbon spectra, in the spectrum of multiblock copolymer, some linkages of **DDP** and **PPD** (as shown in **Q12**) have been observed besides the signal of PDMA and PPOMT.

6. Line 158: 62 DMA and 72 POMT: how is it calculated/estimated?

Reply: The chain length is estimated based on the initial feed ratio and the final conversions of POMT and DMA. The initial feed ratio was $[\text{BDMAT}]:[\text{POMT}]:[\text{DMA}] = 1:100:100$ and after successive heating and irradiation the conversion of POMT and DMA reached 72% and 62%, respectively. Thus, we calculated that average 62 DMA units and 72 POMT units were incorporated in one polymer chain. It is noted that the proton signal of trithiocarbonate (BDMAT) is covered by proton signal of PDMA in ^1H NMR spectrum of isolated polymer,

so it is difficult to using NMR spectrum of isolated polymer to calculate the average chain length. As we know, trithiocarbonate-based RAFT process can efficiently control the radical polymerization of vinyl monomers, and our experiment also demonstrated the ability of BDMAT to control the ring-opening polymerization of POMT. Furthermore, theoretical molecular weight (18.3 kDa) calculated by the initial feed ratio and the final conversions was similar to SEC value (18.0 kDa), so we think chain length estimated based on the initial feed ratio and the final conversions is appropriate.

7. Figure 3d: how to explain the PDI values decreasing with higher conversion and formation of block copolymers?

Reply: For RAFT polymerization, $M_w/M_n = 1 + 1/X_n + 8/[3(k_{dt})]$, (Macromolecules 2001, 34, 40; J. Polym. Sci., Part A: Polym. Chem. 1999, 37,1885), and so, the PDI values decrease with molecular weight or monomer conversion or polymerization time. On the hand, TPPCl catalyzed AROP of cyclic thirane-typed monomers is also living polymerization, in which PDI values decrease with higher molecular weight (Macromolecules 2007, 40, 8129). For the formation of block copolymer, under irradiation, PET-RAFT polymerization produces PDMA-S-C(=S)-S-PDMA, on removing light and heating the system, PDMA-S-C(=S)-S-PDMA could act as a macroinitiator for AROP of POMT, and POMT inserted into the polymer chain between, forming triblock copolymer PDMA-b-PPOMT-S-C(=S)-S-PPOMT-b-PDMA. The detailed mechanism was shown in **Supplementary Figure 1**.

8. Test reaction: after a reversible switch, what happen in the absence of both light and heat? Does the polymerisation completely stop? is it possible to restart ?

Reply: In our experiments, the polymerizations in dimethylacetamide (DMAc) or DMSO can completely stop many times in the absence of both light and heat. And they could also restart several times in the presence of stimuli. We added supplementary experiment in our revised manuscript (**Supplementary Figure 10**), in which prolonged intervals (3 hours) in the absence of both light and heat were operated during polymerization process. No conversion was observed in the absence of both light and heat.

Conversion(POMT) = 40%; Conversion(DMA) = 0%

Conversion(POMT) = 40%; Conversion(DMA) = 0%

Conversion(POMT) = 40%; Conversion(DMA) = 24%

Conversion(POMT) = 40%; Conversion(DMA) = 24%

Conversion(POMT) = 65%; Conversion(DMA) = 24%

¹H NMR trace of the polymerization process containing the absence of both light and heat.

9. Figure 4f: PDI values are missing

Reply: We have added the PDI data to Figure 4f.

10. Line 267: new monomers are tested but the orthogonality has not been proved (as it has been done with DMA and POMT)

Reply: Thanks for the referee's valuable correction. We have added additional experiment to prove the orthogonality of MT and NIPAM during our polymerization process (**Supplementary Figure 11**). And we have also added some necessary statement in our revised manuscript as "the results demonstrated that only MT was polymerized with heat and only NIPAM was polymerized with blue light. Furthermore, there was no any consumption of both MT and NIPAM without both light and heat, confirming the orthogonality of MT and NIPAM (**Supplementary Figure 11**)".

11. Line 268-273: Clearly annotated NMR spectra are needing.

Reply: Thank the referee very much. We have added annotated NMR spectra in our revised manuscript as **Supplementary Figure 12**).

12. How different are the NMR spectra corresponding to block copolymer to gradient polymers (obtained by light and heat at the same time). New linkage should be observable by NMR.

Reply: Thank the referee for the very valuable suggest. There are some differences in NMR spectra between multiblock copolymer and gradient copolymer. In ^{13}C NMR spectrum of multiblock copolymer, some linkages of **DDP** and **PPD** (as shown in the follows) have been observed besides the signal of PDMA and PPOMT. The new linkages (such as **PDP** and **DDP**) beside **DDP** and **PPD** of the gradient copolymers were observed in ^{13}C NMR spectrum as follows:

Reviewer #2:

This is an excellent paper. The experimental data support the conclusion of this paper.

Although there are few examples of orthogonal polymerizations, see ROP and PET-RAFT was perhaps the first one using two different wavelength (Chemical Communications 52, 7126-7129), this paper demonstrates two different stimuli, temperature and light. Both stimuli

appears orthogonal. In my opinion, this paper meets the criteria for publication in Nat. Comm.

1. The authors should comment and explain what is the difference between reference 30 and this one. Indeed, ref 30 demonstrates two orthogonal polymerizations controlled by 2 different wavelengths. This is relatively close to this work. As a consequence, the authors should clearly describe this previous work.

Reply: We thank the Referee. In Ref 30, the authors reported a novel and interesting method to prepare block copolymers by two orthogonal polymerizations using different wavelengths. The authors used a bifunctional initiator to initiate the acrylate and the valerolactone polymerization, containing both trithiocarbonate and terminal hydroxyl group. Trithiocarbonate induced RAFT process while terminal hydroxyl group induced ring-opening polymerization of valerolactone under orthogonal wavelengths respectively to produce diblock copolymer. The method is very useful for prepare di or triblock copolymer, but it is not suitable for the synthesis of polymers with on-demand sequence structure. In this work, we demonstrated not only a dual switchable polymerization under orthogonal stimuli, but also one interconvertible process. Because trithiocarbonate serves as both chain transfer agent (for PET-RAFT) and initiator (AROP), the radical polymerization and ring-opening polymerization could be interconvertible by trithiocarbonate. Thus, under orthogonal stimuli, block, multiblock, gradient and other polymers with on-demand sequence structures could be easily produced. According to Referee's comment, we have clearly described the previous work of Ref 30 in the introduction part of revised manuscript as “**Boyer reported a novel and interesting method to prepare block copolymer by two orthogonal polymerizations using different wavelengths**” in the introduction.

2. I recommend to include NMR data in the main text. Indeed, it is important information to provide to the readers to demonstrate that both monomers are copolymerized under a specific stimuli.

Reply: Thank the referee for the very helpful suggest. We have moved some NMR spectra (such as **Supplementary Figure 4 and Supplementary Figure 6**) into text as **Figure 2e and Figure 3e**.

3. The authors provide numerous schemes in the text. It is obviously good to illustrate the text with schemes, but in my opinion, too many schemes are used. I recommend to incorporate more experimental data, i.e. SEC traces, and NMR to support these conclusions.

Reply: Thank the referee for the very helpful suggest. We have moved some NMR spectra (**Supplementary Figure 4 and Supplementary Figure 6**) into text as **Figure 2e and Figure 3e**; SEC traces (**Supplementary Figure 8, Supplementary Figure 10, Supplementary Figure 17, and Supplementary Figure 20**) into the revised text as **Figure 4 c, Figure 4f, Figure 6c, and Figure 6f**, respectively; the plots of monomer conversation vs polymerization time (**Supplementary Figure 16, Supplementary Figure 19**) to the text as **Figure 6b and Figure 6e**.

4. The concept to use temperature to manipulate the polymer structure has also been proposed in this paper: Polymer Chemistry 8 (39), 6024-6027.

Reply: Thank the Referee very much for the advice. We have cited this paper as Ref 44. as “Boyer and Zhu have proposed a concept to use temperature to manipulate the polymer structure”.

5. Some sentences need to be improved.

Reply: We have improved the writing in text.

6. DSC experiments, I fully agree with the authors that one Tg could mean two monomers are copolymerized. can the authors add a control experiment showing that if these copolymerize in sequence, i.e. prepare block copolymers, two Tg will be observed.

Reply: We thank the Referee for this valuable suggestion. We have run DSC measure of ABA triblock copolymer. The data have been added in our manuscript (as **Supplementary Figure 9c**). We have added “In DSC curve of ABA triblock polymer, two Tg are observed, demonstrating microphase separation of long blocks in resulting copolymer” into the revised text.

7. Perhaps, the author could be interested to cite this paper: Chemical Science 6 (2), 1341-1349 in the introduction showing that PET-RAFT process can use a wide range of catalysts.

Reply: We thank the Referee very much for the advice to perfect our reference citations. We have added this paper (Chemical Science 2015, 6 (2), 1341-1349) as **Ref 58** in our revised manuscript.

Reviewer #3:

This is a nice paper describing what in essence is a very simple idea which is reactions that occur at elevated temperatures will have a very slow rate at ambient temperature. Reactions that require light can usually occur at ambient temperature. Thus simply by lowering the temperature reactions that require 60-90C can pretty much be stopped whilst light is applied. Having said that I am unaware of anyone else who has applied this simple premise and thus this is simplicity is the papers main aspect. They have chosen a rather esoteric AROP monomer rather than a more simple reaction but this is obviously something they are familiar with so why not. They go on to use this to make a multiblock copolymer and show ABA can be made BAB. There is a question over the applications of multi-block copolymers and so it is good to see the collection of thermal data as a measure of properties. However, this is the

least convincing of the data and I would like to see maybe DMA or a better measure of T_g prior to acceptance.

Reply: We thank the Referee very much for the valuable comments. We have run DMA on some samples, the results are similar to those of DSC. The ABA triblock copolymer has two T_g s, while there is only one T_g for multiblock copolymer. There is only small difference of T_g values between DSC and DMA methods.

DMA curve of multiblock copolymer.

DMA curve of ABA triblock copolymer.

It is good to see the GPC traces but I would like to see the data collated into a table. Also we see considerable broadening of the dispersity. What is the dispersity of the polymers shown in suppl figure 8 for example. We are told in the text the curves are all unimodal but they look quite broad so how do the authors know this? Then maybe a discussion of why dispersity seems to broaden as blocks are added.

Reply: We have added all values (including Mn and PDI) into SEC Figures into text in our revised manuscript (Figure 4c, 4f, 4i). The PDIs are about 1.2-1.5. After three cycles of heat and irradiation, the obtained undecablock polymer had the PDI of 1.5 measured by SEC, which was slightly broader than ABA and BAB triblock polymers (~1.4). The AROP is highly temperature dependent. The slightly broadening dispersity may result from that polymerization temperatures can not be decreased to 20 °C from 60 °C immediately, there was some delay on the temperature decrease. Now we have optimized the conditions, and can prepare block copolymer with narrow PDI.

REVIEWERS' COMMENTS:

Reviewer #1 (Remarks to the Author):

The authors thoroughly addressed the comments. The suggested experiments/analysis have been carried out and the results support the discussion. The manuscript has been updated accordingly and meet the criteria for publication in Nature Communication.

Reviewer #2 (Remarks to the Author):

The authors have answered all my questions

Reviewer #3 (Remarks to the Author):

I think the authors have done an excellent job in answering the reviewers questions. The manuscript is now in a very good state and suitable for publication in nature Communications.

I look forward to seeing it published.